# The impact of smartphone use on gait in young adults: Cognitive load vs posture of texting

**Sung-Hyeon Kim**[1]ᵒ, **Jin-Hwa Jung**[2]ᵒ, **Ho-jin Shin**[1], **Suk-Chan Hahm**[3]*, **Hwi-young Cho**[4]*

**1** Department of Health Science, Gachon University Graduate School, Incheon, Republic of Korea, **2** Department of Occupational Therapy, Semyung University, Jecheon, Republic of Korea, **3** Graduate School of Integrative Medicine, CHA University, Seongnam, Republic of Korea, **4** Department of Physical Therapy, Gachon University, Incheon, Republic of Korea

ᵒ These authors contributed equally to this work.
* skchanhahmu@gmail.com (SCH); hwiyoung@gachon.ac.kr (HYC)

**Data Availability Statement:** All relevant data are within the paper and its Supporting Information files.

**Funding:** JH Jung National Research Foundation of Korea (NRF-2018R1C1B5041760) https://www.nrf.

## Abstract

Many researches have reported that the use of smartphones has a negative impact on gait variability and speed of pedestrians by dispersion of cognition, but the influence of factors other than cognitive function on gait is still unclear. The purpose of this study was to investigate the impact of smartphone use on spatiotemporal gait parameters in healthy young people while walking. 42 healthy young adults were recruited and instructed to walk in four conditions (walking without using a smartphone, typing on a smartphone with both hands, typing on a smartphone with one hand, and texting posture with non-task). All spatiotemporal gait parameters were measured using the GAITRite walkway. Compared to walking without using a smartphone, the subjects walked with a slower cadence and velocity and changed stride length and gait cycle and spent more time in contact with the ground when using a smartphone (p < 0.05). In addition, even if a texting posture was taken without performing a task, a similar change was observed when using a smartphone (p < 0.05). This study found that a cautious gait pattern occurred due to smartphone use, and that a change in gait appeared just by taking a posture without using smartphone.

## Introduction

Smartphones are one of the most essential items in the daily life of modern people. In 2018, the smartphone penetration rate exceeded 70% in more than 9 countries, including the United States and the United Kingdom, and more than 250 million people in the United States are using smartphones. Korea's smartphone penetration rate is 11th in the world, and more than 34 million people are using it [1, 2]. Along with the rapid spread of smartphones, the problems caused by the use of smartphones are also increasing.

The use of smartphones while walking reduces the situational awareness and distracts attention and work memory, negatively affecting visual information on the road, motor control and

re.kr/index The funders had no role in study design, data collection and analysis, decision to publish, or preparation of the manuscript.

**Competing interests:** The authors have declared that no competing interests exist.

response [3–9]. The reduction in executive functions due to these dual tasks can affect your ability to walk efficiently and safely [10]. In particular, a change in the walking pattern may increase due to an increase the prioritization of smartphone use among dual tasks [11]. In addition to the dispersion of cognition caused by smartphone use, some studies have reported that smartphone use deforms posture and change gait patterns. Schabrun et al. [12] reported that smartphone use affects gait by reducing the trunk movement of pedestrians, and Jeon et al. [13] reported that there are differences in gait patterns according to the difficulty of tasks using smartphones. In addition, blocking of visual information in the lower visual field by the smartphone and upper limbs may also affect gait [14]. However, the analysis of the various factors that affect the use of smartphones on gait is insufficient. Therefore, analysis of various spatiotemporal gait parameters is required to clearly understand the impact of smartphone use on gait.

The purpose of this study is to investigate the impacts of changes in cognition and posture due to the smartphone use during walking on spatiotemporal gait parameters of healthy young people.

## Materials and methods

### Subjects

This study was conducted in a laboratory setting in Gachon University. The data were collected from August 26 to September 20, 2019. Subjects were recruited through an advertisement and poster on a bulletin board. The inclusion criteria for this study were: (1) Healthy young adult (20–29 years) with no abnormalities in visual and auditory function, musculoskeletal and nervous system, (2) Using a smartphone for at least 6 months, and (3) Can type a smartphone with one or both hands. The exclusion criteria for the study were: 1) Those who have experienced accidents due to smartphone use; 2) Those who are unfamiliar with the use of a QWERTY virtual keyboard; 3) Those who have difficulty typing in English or do not know the English alphabet or numbers; and 4) Those who have problems with normal walking due to fatigue, dizziness, and damaged musculoskeletal system of the lower limbs. The 62 subjects who indicated their willingness to participate in the study received a full explanation about this study before signing a consent form. Volunteers performed a screening after signing the consent form and those who did not meet the inclusion criteria after the research physiotherapist's screening were excluded from the study. Finally, 42 healthy young adults participated in this study. This study was approved by the Institutional Review Board of the Gachon University (1044396-201907-HR-126-01), and enrolled in the Clinical Research Information Service that complies with the World Health Organization International Clinical Trials Registry Platform (WHO-ICTRP) (registration number: KCT0004875). All procedures were done in compliance with the Declaration of Helsinki. The individual in this manuscript has given written informed consent (as outlined in PLOS consent form) to publish these case details.

### Study design

This study was conducted with a single group repeated measure design. All measurements were taken on the same day. All subjects underwent anthropometric measurements (body mass, height, and lower limb lengths) prior to participation. All subjects were instructed to walk in 4 conditions (walking without using a smartphone (baseline, BL), typing on a smartphone with both hands (TSBH), typing on a smartphone with one hand (TSOH), and texting posture with non-task (TPNT)), and the walking order was randomly assigned by the counter balancing method. In addition, the assignment of number to be input into the smartphone during repeated measurement was randomized. All data collection and statistical analysis were

performed by researcher with at least a master's degree blinded to the study method and purpose, respectively. All research process were conducted under the supervision of a physical therapist with at least a master's degree and at least five years of clinical experience.

## Measurement

For measurement, the GAITRite walkway (CIR systems Inc, USA) with excellent test-retest reliability (0.90–0.97) and inter-rater reliability (0.94–0.99) in measuring spatiotemporal gait parameters was used [15]. The instrument is a 180 x 35.5 x 0.25-inch (length x width x height) mat with 13,776 pressure sensors embedded in the surface at intervals of 1.27 cm, and the mat's active area is 144 x 24 inches. The sampling rate was set to 100 Hz, and the measured data was collected and processed using a laptop connected through a serial port.

## Gait data collection procedure

The gait parameters were measured in a visually and acoustically unobstructed corridor environment that was more than 20 m in length and 2 m in width [16, 17]. The subjects were instructed to walk in four experimental conditions (BL, TSBH, TSOH, TPNT) (Fig 1). In all the tasks of texting, the subjects were asked to type in English using the QWERTY virtual keyboard with their smartphone. They were instructed to enter in ascending or descending direction, starting with a random number between 0 and 20. The ascending or descending direction and random number were determined using the website randomization.com (http://www.randomization.com). They were instructed to walk preferred speed at all times while texting without making a mistake. In the TSBH condition, they were instructed to use a smartphone with both hands, and position the smartphone with a comfortable posture and at a comfortable hand height for typing. In the TSOH condition, they were instructed to use the smartphone with their dominant hand and place the smartphone in a comfortable position for texting. The other arm was ordered to swing or lower naturally. To ensure that the TPNT condition is in the same posture as the TSBH condition, they were instructed to type numbers for 5 seconds while standing on the starting line before starting to walk and then turn off the screen and walk while maintaining the posture. To exclude the cognition burden, TPNT condition is designed. TPNT condition is to reproduce the same attitude as using a smartphone without using the smartphone. TSBH and TSOH condition are cognitive dual tasks where both walking and texting tasks are performed simultaneously, and TPNT condition is a single task that walks without texting tasks. To eliminate the learning effect due to repeated

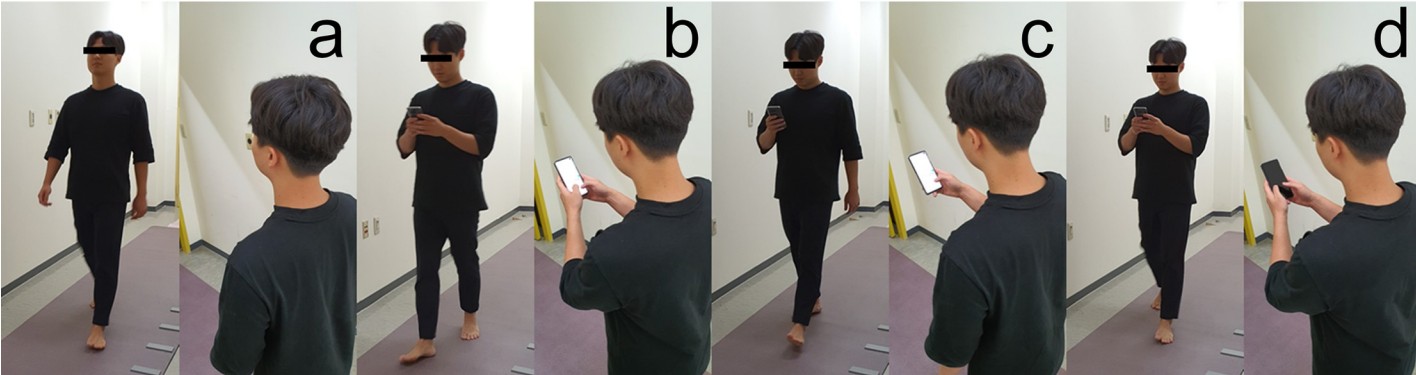

**Fig 1. Illustration of the experimental conditions of the "use of smartphones while walking".** a, walking without using a smartphone; b, typing on a smartphone with both hands; c, typing on a smartphone with one hand; d, texting posture without performing a task.

measurements and to let them become familiarized with test typing and smartphone use posture, they were instructed to walk back and forth on the GAITRite walkway while repeating each condition for five minutes prior to measurement. All conditions were randomly assigned to subjects by the complete counterbalancing method to eliminate order effects due to repeated measurements. All subjects were instructed to walk on the GAITRite walkway with their bare feet for standardize the condition of the feet of all subject [18]. To prevent overlap and potential fatigue effects between measurements, they were instructed to take a 1-minute break between each repeated section and 5-minute break between conditions. They were instructed to start walking from the starting line 2 m before the end of GAITRite and finish walking at the stop line 2 m behind the end of the mat to exclude the acceleration and deceleration section from the gait cycle. When walking on the GAITRite walkway, they were instructed not to step on either end of the mat, in which pressure sensors were not distributed, or to step off the mat. When they stepped off the mat or stopped walking, measurements were taken again. All conditions were repeated three times for measurement. The collected data were processed using a gait analysis software program (GAITRite Plus version 4.7, CIR, system Inc). All data were classified into three categories (rhythm, pace and phases) according to the characteristics of the parameters [19].

## Statistical analyses

The statistical analysis in this study was conducted using SPSS 25.0 software (SPSS Inc, Illinois, USA). All the result parameters used were calculated by mean and standard deviation obtained by measuring each condition three times and the normality of raw data was tested by Shapiro-Wilk test. All parameters following normality were compared between each conditions by repeated measures analysis of variance (ANOVA), and the significance level of the post-hoc test was corrected by the Bonferroni correction. All parameters that did not follow normality were compared between conditions by Friedman test, and the significance level of the post hoc test was corrected by Wilcoxon signed-rank test. For the parameters divided into left and right, the differences between left and right were compared by paired t-test or Wilcoxon signed-rank test. Statistical significance level was set to P <0.05.

# Results

## General characteristics

Fig 2 shows the flow of the subjects who participated in the experiment. Of the 42 subjects in this study, 14.3% (n = 6) of the subjects (gender, male/female: 3:3, age: 24.67 ± 2.58 yrs, height: 167.42 ± 8.05 cm, weight: 63.72 ± 14.92 kg, leg length left: 85.58 ± 5.44 cm, leg length right: 85.75 ± 5.21) dropped-out and the remaining 85.7% (n = 36) completed the evaluation. When an extraneous variable that might affect spatiotemporal gait parameters occurred (complains of awkwardness in maintaining a natural barefoot gait in an experimental environment despite undergoing a sufficient familiarization process (n = 5), abnormal data due to apparatus error (n = 1)), it was considered to removed. Thirty-six subjects except dropout subjects went through repeated measurement, and the characteristics of the subjects are shown in Table 1.

## Changes in spatiotemporal gait parameters due to smartphone use

Changes in spatiotemporal gait parameters due to smartphone use are shown in Table 2. Compared with the BL condition, a significant difference was observed in all parameters of the TSBH condition and TSOH condition (Table 3) (P <0.05). Compared to the BL condition, a significant difference was observed in all parameters except for swing time and single support time (P <0.05).

# Subject flow diagram

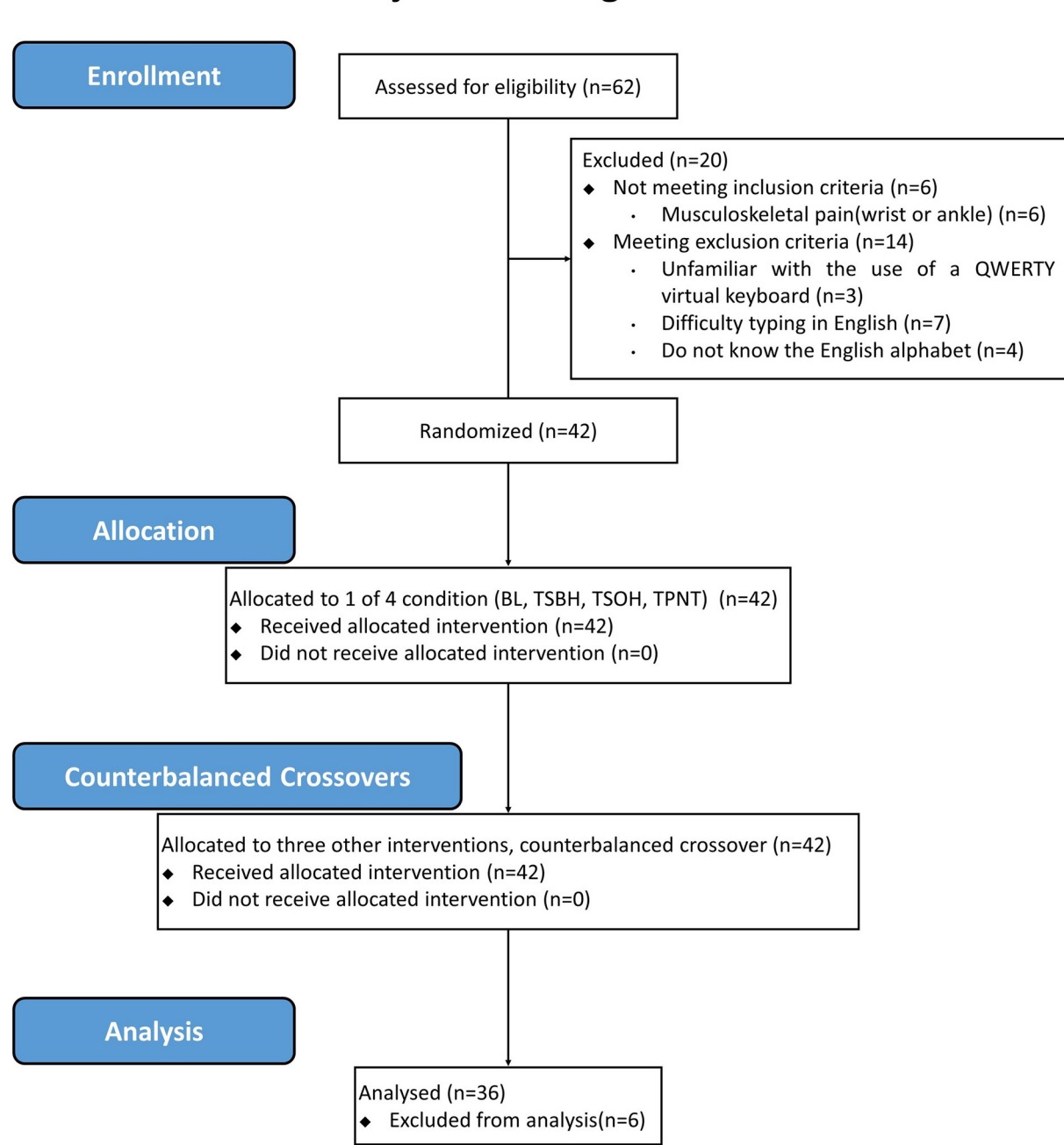

**Fig 2. Subject flow diagram.** BL, walking without using a smartphone; TSBH, typing on a smartphone with both hands; TSOH, typing on a smartphone with one hand; TPNT, texting posture without performing a task.

## Changes according to smartphone use mode and whether task was performed

Compared with the TSBH condition, the differences in the amount of change in spatiotemporal gait parameters while walking under the TSOH condition and the TPNT condition are shown in Table 3. There was no significant difference in all parameters between the TSBH

**Table 1. General characteristics of the subjects.**

| Parameters | | Value (N = 36) |
|---|---|---|
| Gender, male/female (n) [*] | | 26 / 10 |
| Age (year)[†] | | 24.69 ± 1.94 |
| Height (cm)[†] | | 172.11 ± 8.39 |
| Weight (kg)[†] | | 65.66 ± 12.56 |
| Leg length (cm)[†] | Lt. | 87.26 ± 5.14 |
| | Rt. | 87.46 ± 5.09 |

Lt. = Left side; Rt. = Right side.

[*] Values are expressed as number (n).

[†] Values are expressed as mean + SD.

condition and TSOH condition. TPNT condition had a significant difference in gait speed, cadence, step length, step extremity ratio, stride length, swing time, stance time, single support time, double support time, and normalized velocity compared to TSBH and TSOH ($P < 0.05$), and no significant difference in swing during cycle, stance during cycle, single support cycle, and double support cycle.

## Discussion

The main purpose of this study was to investigate changes in spatiotemporal gait parameters caused by smartphone use. The results of this study demonstrated that changes in posture as well as changes in cognitive function due to the smartphone use affect walking.

Previous studies have reported that the impact of cognitive function due to the smartphone use during walking changes gait parameters. The results of this study (gait speed: -21.53%,

**Table 2. The difference between the spatiotemporal gait parameters of four walking conditions.**

| Outcome measure | BL | TSBH | TSOH | TPNT | p |
|---|---|---|---|---|---|
| Rhythm | | | | | |
| Cadence (steps/min) | 115.47 ± 8.03 | 104.78 ± 10.46 | 105.63 ± 10.45 | 112.11 ± 8.64 | < 0.001 |
| Swing time (s) | 0.41 ± 0.02 | 0.44 ± 0.04 | 0.44 ± 0.04 | 0.42 ± 0.03 | < 0.001 |
| Stance time (s) | 0.63 ± 0.06 | 0.72 ± 0.09 | 0.71 ± 0.09 | 0.66 ± 0.06 | < 0.001 |
| Single support time (s) | 0.41 ± 0.02 | 0.44 ± 0.04 | 0.44 ± 0.04 | 0.42 ± 0.02 | < 0.001 |
| Double support time (s) | 0.22 ± 0.04 | 0.28 ± 0.06 | 0.28 ± 0.06 | 0.24 ± 0.04 | < 0.001 |
| Phases | | | | | |
| Swing (%GC) | 39.46 ± 1.17 | 38.11 ± 1.22 | 38.09 ± 1.47 | 38.61 ± 1.24 | < 0.001 |
| Stance (%GC) | 60.54 ± 1.19 | 61.91 ± 1.23 | 61.92 ± 1.48 | 61.40 ± 1.23 | < 0.001 |
| Single support (%GC) | 39.68 ± 1.46 | 38.24 ± 1.63 | 38.07 ± 1.55 | 38.69 ± 1.50 | < 0.001 |
| Double support (%GC) | 20.84 ± 2.28 | 23.77 ± 2.44 | 23.71 ± 2.45 | 22.64 ± 2.36 | < 0.001 |
| Pace | | | | | |
| Gait speed (cm/s) | 129.78 ± 18.33 | 101.84 ± 15.84 | 102.88 ± 18.40 | 116.58 ± 16.68 | < 0.001 |
| Step length (cm) | 67.23 ± 6.66 | 57.63 ± 5.55 | 57.74 ± 6.68 | 61.96 ± 5.95 | < 0.001 |
| Stride length (cm) | 134.63 ± 13.06 | 116.34 ± 10.21 | 116.29 ± 12.69 | 124.52 ± 11.38 | < 0.001 |
| Normalized velocity | 1.49 ± 0.20 | 1.17 ± 0.19 | 1.18 ± 0.20 | 1.34 ± 0.19 | < 0.001 |
| Step extremity ratio | 0.77 ± 0.07 | 0.66 ± 0.07 | 0.66 ± 0.08 | 0.71 ± 0.07 | < 0.001 |

Values are expressed as mean ± SD. *p*-values were determined by the repeated-measures analysis of variance. BL, walking without using a smartphone. TSBH, typing on a smartphone with both hands. TSOH, typing on a smartphone with one hand. TPNT, texting posture with non-task. cm, centimeters. s, seconds. %GC, % gait cycle.

**Table 3. The post-hoc comparison of the spatiotemporal gait parameters.**

| Outcome measure | BL—TSBH | BL—TSOH | BL—TPNT | TSBH—TSOH | TSBH—TPNT | TSOH—TPNT | ANOVA-*p* |
|---|---|---|---|---|---|---|---|
| Rhythm | | | | | | | |
| Cadence (steps/min) | -10.69*** | -9.84*** | -3.36** | 0.85 | 7.33*** | 6.48*** | < 0.001 |
| Swing time (s) | 0.03*** | 0.02*** | 0.00 | 0.00 | -0.02*** | -0.02*** | < 0.001 |
| Stance time (s) | 0.08*** | 0.08*** | 0.03**** | -0.01 | -0.06*** | -0.05*** | < 0.001 |
| Single support time (s) | 0.03*** | 0.02*** | 0.00 | -0.01 | -0.03*** | -0.02*** | < 0.001 |
| Double support time (s) | 0.06*** | 0.06*** | 0.02*** | 0.00 | -0.04*** | -0.03*** | < 0.001 |
| Phases | | | | | | | |
| Swing (%GC) | -1.35*** | -1.37*** | -0.85** | -0.01 | 0.50 | 0.52 | < 0.001 |
| Stance (%GC) | 1.38*** | 1.38*** | 0.86** | 0.01 | -0.52 | -0.53 | < 0.001 |
| Single support (%GC) | -1.44*** | -1.61*** | -0.99*** | -0.17 | 0.46 | 0.63 | < 0.001 |
| Double support (%GC) | 3.11*** | 3.16*** | 1.66*** | 0.05 | -1.45* | -1.50* | < 0.001 |
| Pace | | | | | | | |
| Gait speed (cm/s) | -27.94*** | -26.91*** | -13.20*** | 1.04 | 14.74*** | 13.71*** | < 0.001 |
| Step length (cm) | -9.59*** | -9.49*** | -5.27*** | 0.10 | 4.32*** | 4.22*** | < 0.001 |
| Stride length (cm) | -18.29*** | -18.34*** | -10.12*** | -0.05 | 8.17*** | 8.22*** | < 0.001 |
| Normalized velocity | -0.32*** | -0.31*** | -0.15*** | 0.01 | 0.17*** | 0.15*** | < 0.001 |
| Step extremity ratio | -0.11*** | -0.11*** | -0.06**** | 0.00 | 0.05*** | 0.05*** | < 0.001 |

Values are expressed as difference between means. *Significant difference between each condition by the Bonferroni post-hoc test *, p<0.05. **, p<0.01. ***, p<0.001. ANOVA, analysis of variance. BL, walking without using a smartphone. TSBH, typing on a smartphone with both hands. TSOH, typing on a smartphone with one hand. TPNT, texting posture with non-task. cm, centimeters. s, seconds. %GC, % gait cycle.

cadence: -9.27%, step length: -14.28%, stride length: -13.59%, double support time: + 27.27%) are similar to those of a study by Jeon et al. [13] that reported that the gait difference depends on the difficulty of the task using a smartphone (gait speed: -33.49%, cadence: -17.65%, step length: -19.95%, stride length: -23.14%, double support time: + 81.48%). Our results are also consistent in part with those of a study by Lamberg et al. [20] that reported that smartphone use disperses the cognitive functions of pedestrians, resulting in differences in spatiotemporal gait parameters (gait speed: -33%). It is considered that prior studies showed larger changes because the task (searching and downloading) required a higher level of attention than our study in which a simple task (texting) was conducted. Cognitive function while walking can be distributed due to the increased workload on working memory [21]. Working memory is a multipartite system consisting of a visuospatial sketchpad, episodic buffer, and phonological loop, requiring the maintenance of visual spatial information for a smartphone as well as visual information on the ground during smartphone use. In conclusion, it is considered that smartphone use while walking changes spatiotemporal gait parameters by distributing cognitive functions due to the increased workload on working memory resulting from the increase in visual information as well as attention to performing tasks. In addition, we instructed subjects to type without mistakes in order to focus on the smartphone. These instructions prioritized the smartphone use while the subjects were performing the dual task, and as a result, it seems that it influenced the change of gait parameters [11].

Visual information is important in guiding locomotion by providing feedback and feedforward in performing motor performance [22–25]. In particular, visual information on the lower visual field contributes to correcting the lower limb trajectory or foot placement during walking [22, 26]. However, in the smartphone use posture, the smartphone is fixed toward the front of the field of view, and visual information on the front of the foot is blocked by the device and the upper limb. Decreasing the visual input affects the movement control and can

lead to an instability situation [26]. Our results showed a decrease in gait speed and step length in TPNT as well as TSBH and TSOH. Despite the absence of a cognitive task, the decrease in gait speed and step length in TPNT is a protective adaptation to prevent a dangerous situation due to visual information of a blocked lower visual field, which can be interpreted as a result of a more cautious gait pattern [14, 27]. These results are consistent with the results of Marigold et al. [14], who reported that gait speed and step length decreased in healthy young adults when the lower visual field was reduced. In addition, stance and double support increased under all conditions. It is considered that these change is a compensatory mechanism to increase stability and balance ability by increasing contact time with the ground [28–30].

While walking, which is a high-level movement [10], arm movement is used as one of strategies to increase walking stability, and this affects the maintenance of natural movement of lower limbs [31]. However, during texting, the upper limb is fixed in front of the trunk, which reduces movement, and limited movement reduces natural trunk rotation and lateral flexion, which in turn reduces the ability to walk and to balance [12, 31–33]. Concentrating on a small screen makes the neck and head of users bend forward, reducing their movement. This posture affects the function of the vestibular system, causing navigation errors and restricting walking along a straight track [34]. These points suggest that limited upper limb movements and bent heads may contribute to changes in gait parameters seen in TSBH, TSOH, and TPNT. Interestingly, the same change was observed in the TSOH condition and TSBH condition. These results suggest that a fixed posture of one arm affected the natural movement of the lower limb equally to a fixed posture of both arms.

This study investigated the impact of smartphone use on spatiotemporal gait parameters in healthy adults. However, there are some limitations to this study. First, kinematic changes due to smartphone use could not be measured. As measurements were taken using only GAITRite, it was not possible to accurately measure the angle of the subject's tilted neck and arm swing level during smartphone use. Second, it is difficult to generalize the results of this study in all situations because of the small sample size, and care should be taken when interpreting the results. Third, as this study was conducted in an experimental environment in a space with restricted external intervention, it was not possible to consider various situations that might occur on an actual road. In addition, we performed all measurements with barefoot for standardize the condition of the feet of all subjects. Therefore, this may be different from the outdoor situation in which shoes are worn. Therefore, it is considered that future studies use an larger sample size and conduct analysis of the kinematic changes of the subject in various environmental settings.

In conclusion, the results of this study indicate that smartphone use causes cautious walking due to the dispersion of cognitive function, reduction of visual information, and limitations of body movement. In addition, changes in gait parameters were observed in all smartphone-related conditions, regardless of whether cognitive tasks were performed or not. Therefore, in order to prevent changes in gait caused by the use of a smartphone, it is necessary to pay attention impact on posture as well as cognitive function.

## Supporting information

**S1 Data.**
(XLSX)

## Author Contributions

**Conceptualization:** Sung-Hyeon Kim, Hwi-young Cho.

**Data curation:** Sung-Hyeon Kim, Jin-Hwa Jung, Ho-jin Shin.

**Formal analysis:** Suk-Chan Hahm.

**Funding acquisition:** Jin-Hwa Jung.

**Methodology:** Sung-Hyeon Kim, Jin-Hwa Jung, Suk-Chan Hahm, Hwi-young Cho.

**Project administration:** Sung-Hyeon Kim, Jin-Hwa Jung, Ho-jin Shin.

**Supervision:** Suk-Chan Hahm, Hwi-young Cho.

**Writing – original draft:** Sung-Hyeon Kim, Suk-Chan Hahm, Hwi-young Cho.

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
