## [Editor Report · Decision Letter 0]

12 Apr 2020

PONE-D-20-10060

The effect of smartphone use on gait in healthy young adults: a randomized, repeated measures, counterbalanced, crossover and single-blind study

PLOS ONE

Dear Prof. Cho,

Thank you for submitting your manuscript to PLOS ONE. After careful consideration, we feel that it has merit but does not fully meet PLOS ONE’s publication criteria as it currently stands. Therefore, we invite you to submit a revised version of the manuscript that addresses the points raised during the review process.

Please see my comments below.

We would appreciate receiving your revised manuscript by May 27 2020 11:59PM. To enhance the reproducibility of your results, we recommend that if applicable you deposit your laboratory protocols in protocols.io, where a protocol can be assigned its own identifier (DOI) such that it can be cited independently in the future. For instructions see: http://journals.plos.org/plosone/s/submission-guidelines#loc-laboratory-protocols

We look forward to receiving your revised manuscript.

Kind regards,

Eric R. Anson

Academic Editor

PLOS ONE

Journal Requirements:

2. Please ensure all data are included in the manuscript, as you indicated in the data availability statement.

3. We note that Figure 1 includes an image of a participant in the study.

Additional Editor Comments (if provided):

After reviewing "The effect of smartphone use on gait in healthy young adults: a randomized, repeated measures, counterbalanced, crossover and single-blind study" there are several areas that will need to be addressed before this manuscript can be sent out for peer review.

The study is presented as a clinical trial, but there is only 1 group and all subjects participated in all aspects, please clarify.

It is not clear how this study evaluated walking safety. These claims appear to be over-stated or unsubstantiated, please clarify.

Lines 45-48 are clumsy, consider removing or significantly re-phrasing since they read as opinions. 39% market penetration is not "essential... [for] daily life of modern people"

Paragraph 2 on phone use during driving should be condensed to a single sentence and integrated into the following paragraph.

Line 79, please clarify whether the screening happened prior to the informed consent.

Line 90 is unclear

Line 105, it is not clear what they were instructed to type.

Line 146, please clarify if these subjects were "removed" by the investigators rather than "dropped out" since dropping out indicates a voluntary choice by the subject.

It is not clear from the design or methods how the present study demonstrates a change in cognitive function or posture, this seems like a missed opportunity. There are several areas with claims regarding cognitive distraction or load and it is not clear how that was evaluated, this must be addressed.

Gait speed and balance are not equivalent. See recent work by John Jeka and others in the field of postural control during walking.

The manuscript is concurrently being evaluated for methodology and statistical design, but the above issues need to be addressed with a primary revision responded to in a point by point manner before this manuscript can be reconsidered for peer review.
---

## [Author Response · Author response to Decision Letter 0]

28 May 2020

The comments received from the Editor and Editorial office are described in the 'Response to Reviewers' among uploaded Attach Files.

Please check the file.

---

## [Decision Letter · Decision Letter 1]

27 Jul 2020

PONE-D-20-10060R1

The effect of smartphone use on gait in healthy young adults: a randomized, repeated measures, counterbalanced, crossover and single-blind study

PLOS ONE

Dear Dr. Cho,

Thank you for submitting your manuscript to PLOS ONE. After careful consideration, we feel that it has merit but does not fully meet PLOS ONE’s publication criteria as it currently stands. Therefore, we invite you to submit a revised version of the manuscript that addresses the points raised during the review process.

We look forward to receiving your revised manuscript.

Kind regards,

Eric R. Anson

Academic Editor

PLOS ONE

Reviewers' comments:

Reviewer's Responses to Questions

**Comments to the Author**

1. If the authors have adequately addressed your comments raised in a previous round of review and you feel that this manuscript is now acceptable for publication, you may indicate that here to bypass the “Comments to the Author” section, enter your conflict of interest statement in the “Confidential to Editor” section, and submit your "Accept" recommendation.

Reviewer #1: (No Response)

Reviewer #2: (No Response)

Reviewer #3: (No Response)

2. Is the manuscript technically sound, and do the data support the conclusions?

Reviewer #1: Yes

Reviewer #2: Yes

Reviewer #3: Yes

3. Has the statistical analysis been performed appropriately and rigorously? 

Reviewer #1: (No Response)

Reviewer #2: Yes

Reviewer #3: Yes

4. Have the authors made all data underlying the findings in their manuscript fully available?

Reviewer #1: (No Response)

Reviewer #2: Yes

Reviewer #3: Yes

5. Is the manuscript presented in an intelligible fashion and written in standard English?

Reviewer #1: (No Response)

Reviewer #2: Yes

Reviewer #3: Yes

6. Review Comments to the Author

Reviewer #1: The authors recruited 36 healthy young adults to investigate the effect of smartphone use on spatiotemporal gait parameters. The results showed that subjects with a smartphone use tend to walk slower and change stride length and gait cycle. They concluded to have a negative effect on walking ability using a smartphone.

1. Abstract. Please include your p-value when report the results to support your statement

2. Line 144. 6 out of 42 subjects dropped-out. As the experiment seems to be short and no-harm, it would be informative to provide the reason for dropout. It seems that line 145 might be the reason for dropout? It’s not completely clear if so.

3. Line 146. About barefoot. Why barefoot gait was chosen as people don’t walk barefoot outside? And how the barefoot results apply to the daily life in reality where people wear shoes?

4. Table 1. It would be informative to report the sample characteristics of the dropout subjects in comparison to those of the subjects being analyzed.

5. Table 2. Please define how the difference was calculated (e.g. BL and BS and so on) in the footnote. Also, the presenting numbers for the difference were %, which seems not really represent difference of two measurement only!

6. Table 2. Define in the footnote what test was used corresponding to the p-value

7. Table 3. Annotate how these value were calculate for difference. Are these really the difference or % change?

8. Table 3. What p-values were reported? It seems that two p-value information were reported: one with * notation and one with numerical p-value. But what tests do they correspond to, especially for numerical one. This should be clearly annotated.

Reviewer #2: The authors performed a study in a cohort of 36 healthy adults analysing the effect of smartphone use on gait parameters. The study was performed in a radomized design with blinded investigators. They performed giat tasks four different conditions: Single task, walking with typing with both hands, walking with typing with one hand and walking with caring the smpartphone without typing.

Gait parameters were collected with a gaitrite system. Significant differences betbasline an all three other conditions were found in gait speed, cadence, step length, step extremity ratio, stride length, swing time, stance time, single

support time, double support time, velocity. All parameteres deteriorated in the dual task conditions. Interestingly the effect of the task without typing on the smartphone lead to the same differences in quantitative gait parameters.

The paper is interesting. nevertheless I have some aspects for further improvement of the manuscript:

Minor points:

line 32, line 56 does smartphone use realy reduces cognitive function or cognitive performance durig the task?

line 37: arm swing is difficilt to measure with a gait rite, I guess the condition "walkign withou a smartphone" is described

line 41 walking ability or walking performance?

introduction: the use of a martphone during walk is a dual tasking situation, this has an effect on performance. several cogntive factor are relevant in DT inclutding cognitive flexibiltiy, executive function, prioritization etc. This should be elaborated in more deepth.

methods:

lines 70-67 the sentences starting with "those" describe particitpants not inclusion criteria

line 79 here 42 participants in the abstract 36

line 105 abbreviation DS is not intuitive

which smartphone was used? the same in all participants? the one, which is used in daily life of each participant?

Age range of the participants should be reported

figure 1 look like the participant did not wore shoes. was this the case for all participants? thi sinformation should be added to the methods section.

statistics

line 142 what was the level of significance after bonferoni correction?

it is not clear from the statisics section, if every condition was compares with each other or only with BL.

line 145 statistical: capital letter

Fig 2: why were 14 participants excluded if they met the inclusion criteria

the resolution seems to be low

Results:

line 152 contaminated factor is an unusaul term, which I do not understand

table 2

what is the reference for the categorization into rhythm, pace and phases

it is not clear to which analysis belongs the last column with p-values

the describtion of the tables does not contain any information about the statsitcs used

Major points:

discussion

the association of Dual task and falls is dicussed. In this context the aspects of stops-walking when taking (Lundin-Olsen et al.) and prioritisation (Hobert et al.) needs to be considered in more detail. So a reduction of gait speed e.g. can also be a maneuver of safty, in order to avoid a dangerous situation and not only a marker of deficiency of dual tasking

the effept of carring the smartphone without typing was surprizing to me. I suggest to put a focus on this aspect

the unique selling points: study design with blinded investigator and randomisation und the posture condition are not emphazised and discussed in deepth.

Reviewer #3: The authors conducted an experiment to examine the effects of smartphone use and its typical posture on spatiotemporal gait parameters. The results show smartphone use and its posture change the gait parameters including reduced cadence, speed and stride length. The manuscript is well written and study was conducted nicely. I have only a few comments.

1. The title is misleading as it looks like a clinical trial. This type of laboratory study is typically not considered a clinical trial. I suggest the following title: “The impact of smartphone use on gait in young adults: Cognitive load vs posture of texting”

2. I recommend use replacing “effects” with “impact” or “influence” throughout the manuscript. Although not completely wrong, “effect” is more suitable for treatment effect.

3. Abbreviations “BL” for baseline, “BS” for both hands on smartphone, “DS” for one hand on smartphone and “PS” for posture of smartphone usage are not easy to remember, especially DS.

4. Table 3. There should be an explanation about the “P” in the table header. Same thing for Table 2.

5. Line 197. Gait parameters are indeed used as an indicator for gait ability or dysfunction. However, the slowing down while using smartphone dose not have the same indication for gait dysfunction. I discourage use of “negatively affect” in this context. If they do not slow down when their attention and view of the environment is limited, they are more likely to collide with objects, misstep and/or fall. The fact that the subjects slowed down when using smartphone is a protective adaptation to reduce the risk. The changes in spatiotemporal gait parameters observed all indicate cautious gait pattern.

7. PLOS authors have the option to publish the peer review history of their article (what does this mean?). If published, this will include your full peer review and any attached files.

Reviewer #1: No

Reviewer #2: No

Reviewer #3: No

---

## [Author Response · Author response to Decision Letter 1]

2 Sep 2020

Manuscript Number: PONE-D-20-10060R1

Manuscript Title: The impact of smartphone use on gait in young adults: Cognitive load vs posture of texting

Type of work: Original Article

Dear Reviewers,

I would like to thank you for providing the opportunity to revise and resubmit the attached manuscript entitled “The impact of smartphone use on gait in young adults: Cognitive load vs posture of texting” for publication in PLoS ONE.

We deeply appreciate the Editorial comments and Reviewers’ helpful comments on our manuscript which we ignored. We agreed with the points addressed by the Reviewers. We provide our responses to the Reviewers’ comments. Please review the attached files.

Comments from Reviewer 1:

1. Abstract. Please include your p-value when report the results to support your statement.

Response: 

Thank you for your comment. We additionally described this point as follow:

In line 37-40: “Compared to walking without using a smartphone, the subjects walked with a slower cadence and velocity and changed stride length and gait cycle and spent more time in contact with the ground when using a smartphone (p < 0.05). In addition, even if a texting posture was taken with non-task, a similar change was observed when using a smartphone (p < 0.05).”

2. Line 144. 6 out of 42 subjects dropped-out. As the experiment seems to be short and no-harm, it would be informative to provide the reason for dropout. It seems that line 145 might be the reason for dropout? It’s not completely clear if so.

Response: 

As pointed out by the reviewer, this is a short and non-harmful study. However, the reason why 6 people dropped out is as follows. First, the subject did not normally experience walking barefoot well, and was ashamed to walk in front of people with bare feet exposed. This resulted in very awkward gait posture and gait pattern. These factors were excluded from the results of this study because they may obscure the study results. In addition, there is case in which the machine has a problem with data transmission while measuring the subject and is dropped out.

In accordance with Reviewer’s suggestion, we additionally described these points as follow

In line 153-156: “When an extraneous variable that might affect spatiotemporal gait parameters occurred (complains of awkwardness in maintaining a natural barefoot gait in an experimental environment despite undergoing a sufficient familiarization process (n = 5), abnormal data due to apparatus error (n = 1)), it was considered to removed.”

3. Line 146. About barefoot. Why barefoot gait was chosen as people don’t walk barefoot outside? And how the barefoot results apply to the daily life in reality where people wear shoes?

Response: 

Reviewer suggested us a good point, and this was what we also considered before designing the study. As pointed out by the Reviewer, in daily life, subjects perform walking and other movements while wearing shoes. Therefore, it is more appropriate for the subject to walk with shoes in order to determine the effect of the smartphone on gait. However, for the following reasons, we instructed subjects to perform barefoot walking.

The shoes worn in daily life were different for each subject. Ex) People who prefer sneakers or running shoes, those who prefer shoes, the difference in midsole height, the weight of the shoes, the difference in overlay material, and various types of shoes outsole materials. We conducted a pilot study because these factors affect the subject's gait. We prepared shoes of the same product in various sizes and conducted a study by having subjects wear them. However, the participants did not adjust to the new shoes and walked unnaturally. The subjects were rather uncomfortable and appealed to us that they were unable to walk in their usual way. We thought it would take a lot of time for the subjects to get used to the shoes provided by the researchers, and this situation was not considered appropriate for the purpose of the study.

Therefore, in order to unify the condition of the feet of the participants most consistently, all gait measurements were performed with barefoot, and the subjects did not complain of any discomfort about walking with barefoot.

In addition, in consideration of the reviewer's good points, we added the following contents to the Materials and methods section and the Discussion section as follow:

In line 121-122: “All subjects were instructed to walk on the GAITRite walkway with their bare feet for standardize the condition of the feet of all subject [18].”

In line 235-237: “In addition, we performed all measurements with barefoot for standardize the condition of the feet of all subjects. Therefore, this may be different from the outdoor situation in which shoes are worn.”

4. Table 1. It would be informative to report the sample characteristics of the dropout subjects in comparison to those of the subjects being analyzed.

Response: 

We acknowledge the reviewer’s recommendation. As per the reviewer’s comment, we added the sample characteristics of the dropout subjects as follows:

In line 150-153: “Fig 2 shows the flow of the subjects who participated in the experiment. Of the 42 subjects in this study, 14.3% (n = 6) of the subjects (gender, male/female: 3:3, age: 24.67 ± 2.58 yrs, height: 167.42 ± 8.05 cm, weight: 63.72 ± 14.92 kg, leg length left: 85.58 ± 5.44 cm, leg length right: 85.75 ± 5.21) dropped-out and the remaining 85.7% (n = 36) completed the evaluation.”

5. Table 2. Please define how the difference was calculated (e.g. BL and BS and so on) in the footnote. Also, the presenting numbers for the difference were %, which seems not really represent difference of two measurement only!

Response: 

Authors would like to thanks the Reviewer for a detailed point. We have modified and corrected Table 2 and 3. Please check the Table 2 & 3.

Also, the footnote were added as follows:

In the Table 3: “Values are expressed as difference between means.”

6. Table 2. Define in the footnote what test was used corresponding to the p-value

Response: 

Thank you for your comment. We have added what you pointed out to the footnote.: 

In the Table 2: “p-values were determined by repeated-measures analysis of variance”.

In the Table 3: “*Significant difference between each condition by the Bonferroni post-hoc test *, p<0.05. **, p<0.01. ***, p<0.001. ANOVA, analysis of variance.”

7. Table 3. Annotate how these value were calculate for difference. Are these really the difference or % change?

Response: 

We acknowledge the Reviewer’s comments, and to make it clear, we revised Table 2 & 3 and marked the difference between means in the footnote as follow: 

In the Table 3: “Values are expressed as difference between means.”

8. Table 3. What p-values were reported? It seems that two p-value information were reported: one with * notation and one with numerical p-value. But what tests do they correspond to, especially for numerical one. This should be clearly annotated.

Response: 

We appreciate this suggestion. Authors performed a post-analysis using the Bonferroni correction and performed comparisons between all conditions. 

As you pointed out, we matched the notations for all significance levels, and revised footnotes as follows:

In the Table 3: “Values are expressed as difference between means. Significant difference between each conditions by the Bonferroni test *, p<0.05. **, p<0.01. ***, p<0.001.”

 

Comments from Reviewer 2:

The authors performed a study in a cohort of 36 healthy adults analysing the effect of smartphone use on gait parameters. The study was performed in a radomized design with blinded investigators. They performed giat tasks four different conditions: Single task, walking with typing with both hands, walking with typing with one hand and walking with caring the smpartphone without typing.

Gait parameters were collected with a gaitrite system. Significant differences betbasline an all three other conditions were found in gait speed, cadence, step length, step extremity ratio, stride length, swing time, stance time, single support time, double support time, velocity. All parameteres deteriorated in the dual task conditions. Interestingly the effect of the task without typing on the smartphone lead to the same differences in quantitative gait parameters.

The paper is interesting. nevertheless I have some aspects for further improvement of the manuscript:

Response: 

First of all, all the authors thank Reviewer for his good evaluation and interest. Reviewer checked and suggested a lot of good points about what we overlooked. We agreed these comments and advice, and have made every effort to revise them well. We hope that our revision will be satisfied with the Reviewer.

1. line 32, line 56 does smartphone use realy reduces cognitive function or cognitive performance durig the task?

Response: 

We acknowledge the Reviewer’s comment. We made a mistake and corrected it as follows:

In line 31-33: “Many researches have reported that the use of smartphones has a negative impact on gait variability and speed of pedestrians by dispersion of cognition, but the influence of factors other than cognitive function on gait is still unclear.”

In line 53-54: “In addition to the dispersion of cognition caused by smartphone use, some studies have reported that smartphone use deforms posture and change gait patterns.”

2. line 37: arm swing is difficilt to measure with a gait rite, I guess the condition "walkign withou a smartphone" is described

Response: 

Thanks very much for your detailed observation. We substitute ‘natural arm swing’ to ‘walking without using a smartphone’ as per the Reviewer’s suggestion. 

In line 37-39: “Compared to walking without using a smartphone, the subjects walked with a slower cadence and velocity and changed stride length and gait cycle and spent more time in contact with the ground when using a smartphone (p < 0.05).” 

3. line 41 walking ability or walking performance?

Response: 

All authors appreciate the points of the reviewers. We reconsidered the results of the study and revised the conclusions as follows:

In line 40-42: “This study found that a cautious gait pattern occurred due to smartphone use, and that a change in gait appeared just by taking a posture without using smartphone.”

4. Introduction: the use of a martphone during walk is a dual tasking situation, this has an effect on performance. several cogntive factor are relevant in DT inclutding cognitive flexibiltiy, executive function, prioritization etc. This should be elaborated in more deepth.

Response: 

What the reviewer points out is that all authors agree. In the first version of the manuscript we submitted, we described the content related to dual tasking. However, at the initial check stage of the Editor, Editor requested us to consolidate and shorten the paragraphs related to this content. Thus, we summarized the description of dual tasking considerably as per the Editor's suggestion, and this led to a lack of presentation of the theoretical content for dual tasking.

Accordingly, we reinforced the contents of dual tasking as follows. If you again point out that the content we reinforced is insufficient or inadequate, we will add more theoretical background for dual tasking.

In line 50-53: “The reduction in executive functions due to these dual tasks can affect your ability to walk efficiently and safely [10]. In particular, a change in the walking pattern may increase due to an increase the prioritization of smartphone use among dual tasks [11].” 

5. lines 70-67 the sentences starting with "those" describe particitpants not inclusion criteria

Response: 

Thank you for pointing out our mistakes. The sentence has been modified as follows:

In line 66-68: “The inclusion criteria for this study were: (1) Healthy young adult (20-29 years) with no abnormalities in visual and auditory function, musculoskeletal and nervous system, (2) Using a smartphone for at least 6 months, and (3) Can type a smartphone with one or both hands.”

6. line 79 here 42 participants in the abstract 36

Response: 

Thank you for your comment. The word was revised as follows:

In line 34-36: “42 healthy young adults were recruited and instructed to walk in four conditions (walking without using a smartphone, typing on a smartphone with both hands, typing on a smartphone with one hand, and texting posture with non-task).”

7. line 105 abbreviation DS is not intuitive which smartphone was used? the same in all participants? the one, which is used in daily life of each participant? Age range of the participants should be reported

Response: 

We appreciate the Reviewer's comment and have modified the abbreviations to represent the posture and condition being measured.

Also, we described our study design as 'a single group repeated measre design' in the Study design section, and did not further describe what you pointed out. We have additionally described the following contents to reflect the reviewer's comments in the manuscript as follow:

In line 66-68: “The inclusion criteria for this study were: (1) Healthy young adult (20-29 years) with no abnormalities in visual and auditory function, musculoskeletal and nervous system, (2) Using a smartphone for at least 6 months, and (3) Can type a smartphone with one or both hands.”

In line 85-88: “All subjects were instructed to walk in 4 conditions (walking without using a smartphone (baseline, BL), typing on a smartphone with both hands (TSBH), typing on a smartphone with one hand (TSOH), and texting posture with non-task (TPNT)), and the walking order was randomly assigned by the counter balancing method.”

In line 103-104: “In all the tasks of texting, the subjects were asked to type in English using the QWERTY virtual keyboard with their smartphone.”

8. figure 1 look like the participant did not wore shoes. was this the case for all participants? This information should be added to the methods section.

Response: 

We instructed all subjects to walk barefoot. Other reviewer asked similar question, and we answer this the same.

All participants wear shoes and perform walking and other movements in daily life. Therefore, it is more appropriate for the subject to walk with shoes in order to determine the effect of the smartphone on gait. However, for the following reasons, we instructed subjects to perform barefoot walking.

First, the shoes worn in daily life were different for each subject. Ex) People who prefer sneakers or running shoes, those who prefer shoes, the difference in midsole height, the weight of the shoes, the difference in overlay material, and various types of shoes outsole materials. We conducted a pilot study because these factors affect the subject's gait. We prepared shoes of the same product in various sizes and conducted a study by having subjects wear them. However, the participants did not adjust to the new shoes and walked unnaturally. The subjects were rather uncomfortable and appealed to us that they were unable to walk in their usual way. It was estimated that subjects would take a significant amount of time to adjust to the shoe, which was considered inappropriate for the study.

Therefore, in order to unify the condition of the feet of the participants most consistently, all gait measurements were performed with barefoot, and the subjects did not complain of any discomfort about walking with barefoot.

In addition, in consideration of the reviewer's good points, we added the following to the Materials and methods section and the Discussion section as follow:

In line 121-122: “All subjects were instructed to walk on the GAITRite walkway with their bare feet for standardize the condition of the feet of all subject [18].”

In line 235-237: “In addition, we performed all measurements with barefoot for standardize the condition of the feet of all subjects. Therefore, this may be different from the outdoor situation in which shoes are worn.”

9. line 142 what was the level of significance after bonferoni correction? it is not clear from the statisics section, if every condition was compares with each other or only with BL.

Response: 

I agree and appreciate the reviewer's careful comments. We performed a post-analysis using the Bonferroni correction, and performed comparisons between all conditions. As per the your pointed-out, we revised the Table 2 and 3, and changed the title and footnotes of the Table 3 as follows:

In the Table 3: “The post-hoc comparison of the spatiotemporal gait parameters.”

In the Table 3: “Values are expressed as difference between means. *Significant difference between each condition by the Bonferroni post-hoc test *, p<0.05. **, p<0.01. ***, p<0.001. ANOVA, analysis of variance.”

10. line 145 statistical: capital letter

Response: 

Thank you for your comment. We substitute ‘statistical’ to ‘Statistical’ as follows:

In line 145-146: “Statistical significance level was set to P <0.05.”

11. Fig 2: why were 14 participants excluded if they met the inclusion criteria the resolution seems to be low

Response: 

Thanks for the good point of the reviewer. We increased the resolution of Fig 2 and also presented the details of 'Meeting exclusion criteria' in more detail.

In the Fig 2: “Meeting exclusion criteria (n=14) / Unfamiliar with the use of a QWERTY virtual keyboard (n=3) / Difficulty typing in English (n=7) / Do not know the English alphabet (n=4)”

12. line 152 contaminated factor is an unusaul term, which I do not understand

Response: 

Thank you for your comment. The word was revised as follows:

In line 153-156: “When an extraneous variable that might affect spatiotemporal gait parameters occurred (complains of awkwardness in maintaining a natural barefoot gait in an experimental environment despite undergoing a sufficient familiarization process (n = 5), abnormal data due to apparatus error (n = 1)), it was considered to removed.”

13. table 2 what is the reference for the categorization into rhythm, pace and phases

Response: 

First of all, I'm happy with the Reviewer's good point. We evaluated and analyzed the spatio-temporal parameter among the 4-5 main categories of gait. Various measurement tools and methods are used all over the world to measure this spatio-temporal parameter, and the most standard tool among them is GAITRite. GAITRite can measure various variables corresponding to spatio-temporal parameters. As the Reviewer know well, each variable in the category has different unit and characteristic. We judged and tried to analyze as many of these variables as possible, and performed analysis by classifying gait parameters into rhythm, pace, and phases, referring to the Hollman et al.'s study. This is described in manuscript as follows:

In line 130-131: “All data were classified into three categories (rhythm, pace and phases) according to the characteristics of the parameters [19].”

14. table 2 it is not clear to which analysis belongs the last column with p-values the describtion of the tables does not contain any information about the statsitcs used

Response: 

Thank you for your advisory observation. Corrected Tables 2 and 3 as reviewers pointed-out.

In addition, the table were added as follows:

In the Table 2: “p-values were determined by the repeated-measures analysis of variance”.

In the Table 3: “Values are expressed as difference between means. *Significant difference between each condition by the Bonferroni post-hoc test *, p<0.05. **, p<0.01. ***, p<0.001. ANOVA, analysis of variance.”

15. the association of Dual task and falls is dicussed. In this context the aspects of stops-walking when taking (Lundin-Olsen et al.) and prioritisation (Hobert et al.) needs to be considered in more detail. So a reduction of gait speed e.g. can also be a maneuver of safty, in order to avoid a dangerous situation and not only a marker of deficiency of dual tasking the effept of carring the smartphone without typing was surprizing to me. I suggest to put a focus on this aspect

Response: 

All the authors thank you for the Reviewer’ detailed comment. In order to clarify the effect of smart phone use on gait parameters during walking, we have set up two hypotheses as follows: 

1) As reported in previous studies, smart phone use will affecte gait due to dual tasks. 

2) Not only the influence of cognitive function due to the dual task, but also the change of posture due to the use of a smartphone will affect the change in gait.

As the author pointed out, we have not been able to fully explain the results obtained through the hypothesis in manuscript. We apologize for this. We took the reviewer's advice and added the following contents to the Discussion section. We hope that our additional descriptions will be satisfactory to the Reviewer, and if you think our revised content is insufficient or inappropriate, please let us know.

In line 202-205: “In addition, we instructed subjects to type without mistakes in order to focus on the smartphone. These instructions prioritized the smartphone use while the subjects were performing the dual task, and as a result, it seems that it influenced the change of gait parameters [11].”

In line 206-218: “Visual information is important in guiding locomotion by providing feedback and feedforward in performing motor performance [22-25]. In particular, visual information on the lower visual field contributes to correcting the lower limb trajectory or foot placement during walking [22,26]. However, in the smartphone use posture, the smartphone is fixed toward the front of the field of view, and visual information on the front of the foot is blocked by the device and the upper limb. Decreasing the visual input affects the movement control and can lead to an instability situation [26]. Our results showed a decrease in gait speed and step length in TPNT as well as TSBH and TSOH. Despite the absence of a cognitive task, the decrease in gait speed and step length in TPNT is a protective adaptation to prevent a dangerous situation due to visual information of a blocked lower visual field, which can be interpreted as a result of a more cautious gait pattern [14,27]. These results are consistent with the results of Marigold et al. [14], who reported that gait speed and step length decreased in healthy young adults when the lower visual field was reduced. In addition, stance and double support increased under all conditions. It is considered that these change is a compensatory mechanism to increase stability and balance ability by increasing contact time with the ground [28-30].”

16. the unique selling points: study design with blinded investigator and randomisation und the posture condition are not emphazised and discussed in deepth.

Response: 

Thank you for your careful point. The contents have been revised as follows in order to express the contents well to the Readers. We hope you are satisfied with our additional description about 'blinded investigator', 'randomization', and 'the posture condition'.

In line 85-92: “All subjects were instructed to walk in 4 conditions (walking without using a smartphone (baseline, BL), typing on a smartphone with both hands (TSBH), typing on a smartphone with one hand (TSOH), and texting posture with non-task (TPNT)), and the walking order was randomly assigned by the counter balancing method. In addition, the assignment of number to be input into the smartphone during repeated measurement was randomized. All data collection and statistical analysis were performed by researcher with at least a master's degree blinded to the study method and purpose, respectively. All research process were conducted under the supervision of a physical therapist with at least a master's degree and at least five years of clinical experience.”

 

Comments from Reviewer 3:

1. The title is misleading as it looks like a clinical trial. This type of laboratory study is typically not considered a clinical trial. I suggest the following title: “The impact of smartphone use on gait in young adults: Cognitive load vs posture of texting”

Response: 

We sincerely acknowledge the Reviewer’s pointed-out. The Title you provided is clearly a more appropriate notation, and we have accepted it and amended the Title as follows.

Once again, thank you for your suggestion.

In line 2-3: “The impact of smartphone use on gait in young adults: Cognitive load vs posture of texting”

2. I recommend use replacing “effects” with “impact” or “influence” throughout the manuscript. Although not completely wrong, “effect” is more suitable for treatment effect.

Response: 

Thanks for the good suggestions from the Reviewer. We revised these points throughout the entire manuscript according to the reviewer's comments as follow:

In line 31-34: “Many researches have reported that the use of smartphones has a negative impact on gait variability and speed of pedestrians by dispersion of cognition, but the influence of factors other than cognitive function on gait is still unclear. The purpose of this study was to investigate the impact of smartphone use on spatiotemporal gait parameters in healthy young people while walking.”

In line 58-59: “Therefore, analysis of various spatiotemporal gait parameters is required to clearly understand the impact of smartphone use on gait.” 

In line 60-61: “The purpose of this study is to investigate the impacts of changes in cognition and posture due to the smartphone use during walking on spatiotemporal gait parameters of healthy young people.” 

In line 188-189: “Previous studies have reported that the impact of cognitive function due to the smartphone use during walking changes gait parameters.” 

In line 229: “This study investigated the impacts of smartphone use on spatiotemporal gait parameters in healthy adults.” 

In line 243-244: “Therefore, in order to prevent changes in gait caused by the use of a smartphone, it is necessary to pay attention impact on posture as well as cognitive function.”

3. Abbreviations “BL” for baseline, “BS” for both hands on smartphone, “DS” for one hand on smartphone and “PS” for posture of smartphone usage are not easy to remember, especially DS.

Response: 

Authors would like to thanks the reviewer for the recommendation. All authors accepted this suggestion, and we corrected the abbreviation intuitively as follow:

In line 85-88: “All subjects were instructed to walk in 4 conditions (walking without using a smartphone (baseline, BL), typing on a smartphone with both hands (TSBH), typing on a smartphone with one hand (TSOH), and texting posture with non-task (TPNT)), and the walking order was randomly assigned by the counter balancing method.”

4. Table 3. There should be an explanation about the “P” in the table header. Same thing for Table 2.

Response: 

We appreciate this suggestion. We revised Table 2 & 3 in accordance with the Reviewer's suggestion, and added a description of the "P" in the Footnote. 

In the Table 2: “p-values were determined by repeated-measures analysis of variance”.

In the Table 3: “Values are expressed as difference between means. *Significant difference between each condition by the Bonferroni post-hoc test *, p<0.05. **, p<0.01. ***, p<0.001. ANOVA, analysis of variance.” 

5. Line 197. Gait parameters are indeed used as an indicator for gait ability or dysfunction. However, the slowing down while using smartphone dose not have the same indication for gait dysfunction. I discourage use of “negatively affect” in this context. If they do not slow down when their attention and view of the environment is limited, they are more likely to collide with objects, misstep and/or fall. The fact that the subjects slowed down when using smartphone is a protective adaptation to reduce the risk. The changes in spatiotemporal gait parameters observed all indicate cautious gait pattern.

Response: 

We conducted a study to find out that the cause of the use of smartphones to change gait is not only the change of cognitive function but also the change of posture due to the dual task, and the content was revised to reflect this well in the text. With reference to the content presented by the Reviewer, we discussed what we were trying to present as follows:

In line 202-205: “In addition, we instructed subjects to type without mistakes in order to focus on the smartphone. These instructions prioritized the smartphone use while the subjects were performing the dual task, and as a result, it seems that it influenced the change of gait parameters [11].”

In line 206-218: “Visual information is important in guiding locomotion by providing feedback and feedforward in performing motor performance [22-25]. In particular, visual information on the lower visual field contributes to correcting the lower limb trajectory or foot placement during walking [22,26]. However, in the smartphone use posture, the smartphone is fixed toward the front of the field of view, and visual information on the front of the foot is blocked by the device and the upper limb. Decreasing the visual input affects the movement control and can lead to an instability situation [26]. Our results showed a decrease in gait speed and step length in TPNT as well as TSBH and TSOH. Despite the absence of a cognitive task, the decrease in gait speed and step length in TPNT is a protective adaptation to prevent a dangerous situation due to visual information of a blocked lower visual field, which can be interpreted as a result of a more cautious gait pattern [14,27]. These results are consistent with the results of Marigold et al. [14], who reported that gait speed and step length decreased in healthy young adults when the lower visual field was reduced. In addition, stance and double support increased under all conditions. It is considered that these change is a compensatory mechanism to increase stability and balance ability by increasing contact time with the ground [28-30].”

Added reference:

Response: 

The contents were supplemented according to the Reviewer's advice, and references were added. References added are as follows:

11. Yogev-Seligmann G, Rotem-Galili Y, Dickstein R, Giladi N, Hausdorff JM. Effects of explicit prioritization on dual task walking in patients with Parkinson's disease. Gait Posture. 2012;35(4):641-646. doi:10.1016/j.gaitpost.2011.12.016.

14. Marigold DS, Patla AE. Visual information from the lower visual field is important for walking across multi-surface terrain. Exp Brain Res. 2008;188(1):23-31. doi:10.1007/s00221-008-1335-7.

18. Roche B, Simon AL, Guilmin-Crépon S, et al. Test-retest reliability of an instrumented electronic walkway system (GAITRite) for the measurement of spatio-temporal gait parameters in young patients with Friedreich's ataxia. Gait Posture. 2018;66:45-50. doi:10.1016/j.gaitpost.2018.08.017.

22. Land MF. Eye movements and the control of actions in everyday life. Prog Retin Eye Res. 2006;25(3):296-324. doi:10.1016/j.preteyeres.2006.01.002.

23. Fajen BR, Warren WH. Behavioral dynamics of steering, obstacle avoidance, and route selection. J Exp Psychol Hum Percept Perform. 2003;29(2):343-362. doi:10.1037/0096-1523.29.2.343.

24. Mohagheghi AA, Moraes R, Patla AE. The effects of distant and on-line visual information on the control of approach phase and step over an obstacle during locomotion. Exp Brain Res. 2004;155(4):459-468. doi:10.1007/s00221-003-1751-7.

25. Patla AE, Greig M. Any way you look at it, successful obstacle negotiation needs visually guided on-line foot placement regulation during the approach phase. Neurosci Lett. 2006;397(1-2):110-114. doi:10.1016/j.neulet.2005.12.016.

26. Patla AE, Vickers JN. How far ahead do we look when required to step on specific locations in the travel path during locomotion?. Exp Brain Res. 2003;148(1):133-138. doi:10.1007/s00221-002-1246-y.

27. Helbostad JL, Vereijken B, Hesseberg K, Sletvold O. Altered vision destabilizes gait in older persons. Gait Posture. 2009;30(2):233-238. doi:10.1016/j.gaitpost.2009.05.004.

 

We hope that the revisions we have made are satisfactory. Please inform us if there is anything else that can be done to improve the manuscript.

Sincerely,

Hwi-young Cho, PT, PhD,

Department of Physical Therapy, College of Health Science, Gachon University, 191 Hambangmoe-ro, Yeonsu-gu, Incheon 21936, Republic of Korea 

Tel: +82-32-820-4560; Fax: +82-32-820-4420; E-mail: hwiyoung@gachon.ac.kr

Suk-Chan Hahm, PT, PhD,

Graduate School of Integrative Medicine, CHA University, CHA Bio Complex, 355, Pangyo-ro, Bundang-gu, Seongnam-si, Kyonggi-do, 13488, Republic of Korea

Tel: +82-31-881-7101; Fax: +82-31-881-7069; E-mail: schahm@cha.ac.kr

---

## [Decision Letter · Decision Letter 2]

21 Sep 2020

The impact of smartphone use on gait in young adults: Cognitive load vs posture of texting

PONE-D-20-10060R2

Dear Dr. Cho,

We’re pleased to inform you that your manuscript has been judged scientifically suitable for publication and will be formally accepted for publication once it meets all outstanding technical requirements.

Kind regards,

Eric R. Anson

Academic Editor

PLOS ONE

Additional Editor Comments (optional):

Reviewers' comments:

Reviewer's Responses to Questions

**Comments to the Author**

1. If the authors have adequately addressed your comments raised in a previous round of review and you feel that this manuscript is now acceptable for publication, you may indicate that here to bypass the “Comments to the Author” section, enter your conflict of interest statement in the “Confidential to Editor” section, and submit your "Accept" recommendation.

Reviewer #1: All comments have been addressed

Reviewer #2: All comments have been addressed

Reviewer #3: All comments have been addressed

2. Is the manuscript technically sound, and do the data support the conclusions?

Reviewer #1: (No Response)

Reviewer #2: Yes

Reviewer #3: Yes

3. Has the statistical analysis been performed appropriately and rigorously? 

Reviewer #1: (No Response)

Reviewer #2: Yes

Reviewer #3: Yes

4. Have the authors made all data underlying the findings in their manuscript fully available?

Reviewer #1: (No Response)

Reviewer #2: Yes

Reviewer #3: Yes

5. Is the manuscript presented in an intelligible fashion and written in standard English?

Reviewer #1: (No Response)

Reviewer #2: Yes

Reviewer #3: Yes

6. Review Comments to the Author

Reviewer #1: (No Response)

Reviewer #2: All comments have been adressed satisfactorily and the manuscript has improved.

Reviewer #3: The authors addressed my review comments sufficiently. The writings of this manuscript now appropriately represent what this study investigated and found. I have no further comments.

7. PLOS authors have the option to publish the peer review history of their article (what does this mean?). If published, this will include your full peer review and any attached files.

Reviewer #1: No

Reviewer #2: No

Reviewer #3: No

---

## [Editor Report · Acceptance letter]

2 Oct 2020

PONE-D-20-10060R2 

The impact of smartphone use on gait in young adults: Cognitive load vs posture of texting 

Dear Dr. Cho:

I'm pleased to inform you that your manuscript has been deemed suitable for publication in PLOS ONE. Congratulations! Your manuscript is now with our production department. 

Kind regards, 

on behalf of

Dr. Eric R. Anson 

Academic Editor

PLOS ONE